

**Stratosphere-troposphere separation of nitrogen dioxide**
**columns from the TEMPO geostationary satellite**
**instrument**
**Jeffrey A. Geddes[1,2], Randall V. Martin[1,3], Eric J. Bucsela[4], Chris A. McLinden[5],**
**and Daniel J.M. Cunningham[1]**
[1]Department of Physics and Atmospheric Science, Dalhousie University, Halifax, NS, Canada
[2]Now at the Department of Earth and Environment, Boston University, Boston, MA, USA
[3]Harvard-Smithsonian Center for Astrophysics, Cambridge, Massachusetts, USA
[4]SRI International, Menlo Park, California, USA
[5]Air Quality Research Division, Environment and Climate Change Canada, Toronto, ON,
Canada
*Correspondence to:* J.A. Geddes (jgeddes@bu.edu)
**Abstract**
Separating the stratospheric and tropospheric contributions in satellite retrievals of
atmospheric $NO_2$ column abundance is a crucial step in the interpretation and application of
the satellite observations. A variety of stratosphere-troposphere separation algorithms have
been developed for sun-synchronous instruments in low Earth orbit (LEO) that benefit from
global coverage, including broad clean regions with negligible tropospheric $NO_2$ compared to
stratospheric $NO_2$. These global sun-synchronous algorithms need to be evaluated and refined
for forthcoming geostationary instruments focused on continental regions, which lack this
global context and require hourly estimates of the stratospheric column. Here we develop and
assess a spatial filtering algorithm for the upcoming TEMPO geostationary instrument that
will target North America. Developments include using independent satellite observations to
identify likely locations of tropospheric enhancements, using independent LEO observations
for spatial context, consideration of diurnally-varying partial fields of regard, and a filter
based on stratospheric to tropospheric air mass factor ratios. We test the algorithm with LEO



observations from the OMI instrument with an afternoon overpass, and from the GOME-2
instrument with a morning overpass.

3        We compare our TEMPO field of regard algorithm against an identical global algorithm

to investigate the penalty resulting from the limited spatial coverage in geostationary orbit,
and find excellent agreement in the estimated mean daily tropospheric $NO_2$ column densities
($R^2 = 0.999$, slope = 1.009 for July and $R^2 = 0.998$, slope = 0.999 for January). The algorithm
performs well even when only small parts of the continent are observed by TEMPO. The
algorithm is challenged the most by east coast morning retrievals in the wintertime (e.g. $R^2 =$
0.995, slope = 1.038 at 1400 UTC). We find independent global low Earth observations
(corrected for time of day) provide important context near the field-of-regard edges. We also
test the performance of the TEMPO algorithm without these supporting global observations.
Most of the continent is unaffected ($R^2 = 0.924$ and slope = 0.973 for July and $R^2 = 0.996$ and
slope = 1.008 for January), with 90% of the pixels having differences of less than $\pm\ 0.2\ x\ 10^{15}$
molecules $cm^{-2}$ between the TEMPO tropospheric $NO_2$ column density and the global
algorithm. For near-real-time retrieval, even a climatological estimate of the stratospheric
$NO_2$ surrounding the field of regard would improve this agreement. In general, the additional
penalty of a limited field of regard from TEMPO introduces no more error than normally
expected in most global stratosphere-troposphere separation algorithms. Overall, we conclude
that hourly near-real-time stratosphere-troposphere separation for the retrieval of $NO_2$
tropospheric column densities by the TEMPO geostationary instrument is both feasible and
robust, regardless of the diurnally-varying limited field of regard.

## 23   1   Introduction

24        Nitrogen dioxide ($NO_2$) and nitrogen oxides in general are central to atmospheric

chemistry in both the troposphere and stratosphere (Finlayson-Pitts and Pitts, 1999; Seinfeld
and Pandis, 2016). In the stratosphere, nitrogen oxides are a key player in ozone ($O_3$)
depletion chemistry. In the troposphere, photolysis of $NO_2$ is responsible for the production of
$O_3$ whose buildup is associated with negative human health, ecosystem, and radiative forcing
impacts. Emissions of nitrogen oxides are also linked to the production of secondary
inorganic aerosol with impacts on both health and global climate. Observations of $NO_2$ in the
atmosphere are therefore critical given its roles in air quality and atmospheric chemistry.



Satellite remote sensing of NO₂ from instruments in low Earth orbit has offered
extraordinary insight into global nitrogen oxide processes. Among many applications,
observations from GOME (1996-2003), SCIAMACHY (2002-2011), OMI (2004-), and
GOME-2 (2007-) have contributed to understanding global and regional patterns in nitrogen
oxide emissions (e.g. Beirle et al. 2003; Duncan et al. 2013; Jaegle et al., 2005; Konovalov et
al., 2008; Lamsal et al., 2011; Martin et al., 2003; Miyazaki et al., 2016; Richter et al. 2005;
Russell et al. 2012), evaluating ground-level air quality in the absence of traditional
monitoring data (e.g. Bechle et al., 2013; Boersma et al. 2009; Geddes et al., 2016; Lamsal et
al. 2008; McLinden et al., 2012), and constraining nitrogen oxide deposition out of the
atmosphere (e.g. Geddes and Martin, 2017; Jia et al., 2016; Nowlan et al., 2014). A key step
in these applications is the separation of stratospheric and tropospheric NO₂ from the total
column derived from the satellite observation, a process that can introduce substantial
uncertainty the final tropospheric column estimates (Beirle et al. 2016; Boersma et al., 2004;
Bucsela et al. 2013; Martin et al., 2002).
Separating the stratospheric and tropospheric contributions to the total column has been
performed using a number of approaches, varying in complexity and in the assumptions that
are made. The simplest approach is the Pacific reference sector method (Beirle et al., 2003;
Martin et al., 2002; Richter and Burrows, 2002) in which stratospheric NO₂ is treated as
longitudinally homogeneous so that stratospheric NO₂ in any location can be estimated by
using the measured NO₂ over the remote Pacific at the same latitude. Tropospheric NO₂ in the
reference sector might either be ignored altogether (e.g. Richter and Burrows, 2002) or
accounted for using a model estimate (e.g. Martin et al., 2002). While the treatment of zonal
invariance is reasonable for low- to mid-latitudes, stratospheric dynamics (especially in the
vicinity of polar vortices) raise concerns at higher latitudes of relevance for planned
geostationary missions.
Image processing and spatial filtering techniques are an extension of the reference sector
method (Bucsela et al., 2006, 2013; Leue et al., 2001; Valks et al., 2011; Velders et al., 2001;
Wenig et al., 2004), whereby stratospheric NO₂ is estimated by interpolating between regions
that are classified as having negligible tropospheric NO₂. This might be accomplished for
example by using only cloudy scenes over the oceans (e.g. Leue et al., 2001), or by applying a
pollution "mask" given prior estimates of tropospheric NO₂ (e.g. Bucsela et al., 2006; Valks
et al., 2011). Bucsela et al. (2013) proposed a masking scheme that combines a prior estimate



of tropospheric $NO_2$ with radiative transfer calculations to allow polluted pixels to remain if the scene is cloudy (obscuring lower tropospheric $NO_2$), and exclude unpolluted regions where tropospheric $NO_2$ signal may still be significant due to high tropospheric air mass factors. An elegant variation of this spatial filtering approach is the STRatospheric Estimation Algorithm from Mainz (STREAM), developed by Beirle et al. (2016). Instead of binary masks based on arbitrary thresholds, STREAM applies a weighted convolution scheme where cloudy observations are given a high weight and polluted observations (based on a prior estimate) are given low weight. These spatial filtering approaches developed exclusively for global observational coverage from low Earth orbit offer valuable guidance on the development of geostationary stratosphere-troposphere separation algorithms.

Nadir observations are also used in assimilation approaches where model predictions of the stratospheric $NO_2$ column density are adjusted towards the observed column density. For example, stratosphere-troposphere separation in the Dutch $NO_2$ algorithm is achieved by assimilating observed $NO_2$ columns with model $NO_2$ column predictions from the TM4 chemical transport model forced by ECMWF meteorological data (Boersma et al., 2007; Dirksen et al., 2011). In that approach, modeled $NO_2$ profiles are convolved into line-of-sight ("slant") columns using averaging kernels, and the difference between modeled and observed slant column densities are used to force the modeled columns to an "analysed" state. Using the most recent observations aviable, the "analysed" state can be used in a forecast model run to predict the stratospheric field for near-real time retrievals (Boersma et al. 2007).

In some cases, independent stratospheric observations may be used in the separation of stratospheric and tropospheric $NO_2$. For example, the SCIAMACHY instrument made almost coincident nadir and limb measurements (Bovensmann et al., 1999) and this matching was exploited in algorithms by Beirle et al. (2010) and Hilboll et al. (2013). Even non-coincident limb-nadir matching has been exploited for stratosphere-troposphere separation, as in the case of OSIRIS and OMI (Adams et al., 2016). Sussmann et al. (2005) demonstrate how simultaneous ground-based measurements (especially at mountain sites) could be applied for stratosphere-troposphere separation algorithm validation.

To date, all of the above approaches to stratosphere-troposphere separation have been developed using the large coverage of observations provided by instruments in low Earth orbit. Questions remain about how well the separation can be performed without the global context and where clean tropospheric background signals are limited. Stratosphere-



troposphere separation algorithms need to be evaluated and refined for the restricted field of
regard of future geostationary instruments such as TEMPO (Zoogman et al., 2017), Sentinel-4
(Veihelmann et al. 2015), and GEMS (Lasnik et al. 2014).

4       TEMPO ("Tropospheric Emissions: Monitoring of Pollution"), launching between 2019-

2021, will provide space-based measurements in geostationary orbit with a field of regard
over North America from southern Canada to Mexico City and the Bahamas (Zoogman et al.,
2017). The spectrometer has spectral ranges of 290-490 nm (at 0.57 nm resolution) and 540-
740 nm (at 0.2 nm resolution), allowing retrieval of tropospheric composition with fine spatial
resolution (up to 2.1 km North-South x 4.4 km East-West instantaneous field of view).
Scanning occurs from east to west, with hourly revisits. Among its standard products
available at roughly 4 km x 8 km spatial resolution will be hourly $NO_2$ column abundance.
Here, we develop a standard stratosphere-troposphere separation algorithm for the
observations of $NO_2$ from TEMPO, and examine in detail the potential information penalty
associated with the limited TEMPO field of regard compared to an identical global algorithm.
**2   Satellite Observations**
To develop and test our algorithm, we use data from two low Earth orbiting instruments,
with afternoon and morning overpasses. We use $NO_2$ column densities derived from OMI on
board the Aura satellite launched in 2004. OMI is a nadir-viewing spectrometer in low Earth
orbit crossing the equator around 13:30 local time, with a variable horizontal resolution of 13
km x 24 km at nadir. Line-of-slight ("slant") columns are retrieved from spectral fitting of
back-scattered and reflected solar radiation within the 405-465 nm wavelength range, and
corrected for instrumental artifacts (Bucsela et al., 2013). We use the Version 2.1 Collection 3
data   product   from   NASA   (Krotkov   et   al.   2017,   publicly   available   at
http://disc.sci.gsfc.nasa.gov/Aura/data-holdings/OMI/omno2_v003.shtml),              including
stratospheric and tropospheric air mass factors provided with the data to relate slant and
vertical columns (Bucsela et al., 2013). We use the artifact-corrected slant column densities
("destriping") and the tropospheric and stratospheric air mass factors calculated for each
pixel. All data are first gridded to a 0.1° x 0.1° regular grid.
We also make use of $NO_2$ column densities derived from GOME-2, on board the MetOp-
A satellite launched in 2006. GOME-2 is another nadir-viewing spectrometer in low Earth
orbit, crossing the equator around 09:30 local time with a constant horizontal resolution of 80





km x 40 km in its default swath. Spectral fitting is performed within the 420-450 nm
wavelength range. Here we use the TM4NO2A retrieval (Boersma et al. 2004) version 2.3
data product from KNMI (available from http://www.temis.nl/airpollution/no2.html) along
with the included air mass factors.
We restrict all data to solar zenith angles smaller than 80°.
**3  Estimating Stratospheric NO₂ over the TEMPO Field of Regard**
Here we describe our approach to estimate the stratospheric NO₂ column in TEMPO
observations. As a foundation for our method, we begin with the approach used in the current
operational algorithm for OMI (Bucsela et al., 2013). This algorithm has demonstrated high
quality performance against validation data sets (Ialongo et al., 2016; Lamsal et al., 2014;
Bucsela et al., 2013), is computationally fast, and is suitable for near-real-time retrievals. Our
own implementation of this algorithm reproduces the operational global stratospheric NO₂
product well (r = 0.99 and a slope of 1.01). As described below, we build on this algorithm for
TEMPO by modifying certain smoothing/filtering steps, using a satellite-derived prior
estimate of tropospheric NO₂, incorporating observations surrounding the TEMPO field of
regard from independent low Earth orbit instruments, and by considering partial fields of
regard relevant to TEMPO.
Figure 1 shows the stepwise implementation of our TEMPO stratosphere-troposphere
separation algorithm for an example day in July. As a surrogate for TEMPO observations, we
begin by restricting the OMI total slant NO₂ column observations to the anticipated TEMPO
field of regard below a solar zenith angle threshold of 80° (Figure 1a). The expected coverage
of TEMPO extends from as far south as Mexico City, northward to include southern Canada
(covering as far north as the oil sands region in Alberta for example). The pattern along the
orbit tracks in Figure 1a results from the changing OMI viewing zenith angle (with higher
slant columns for larger viewing angles).
An initial estimate of the stratospheric vertical NO₂ column ($V_{init}$) can be obtained by:
$$V_{init} = \frac{(S - S_{trop,prior})}{A_{strat}}$$           Equation 1
where $S$ is the total slant column density, $A_{strat}$ is the stratospheric air mass factor, and $S_{trop,prior}$
accounts for small contributions from the troposphere (Bucsela et al. 2013). Bucsela et al.



(2013) estimated the tropospheric contribution using model values. To provide a more
accurate constraint on tropospheric contributions, we use the monthly mean tropospheric $NO_2$
columns derived from independent GOME-2 observations as an initial a-priori tropospheric
$NO_2$ estimate. This concept enables the use of spatial information observed from satellite, and
could be readily adapted to use TROPOMI observations at finer resolution. The use of a
satellite-derived a-priori reduces the use of chemical transport model information in the
stratosphere-troposphere separation algorithm (although we revert to a model estimate if
quality controlled satellite coverage is not available, e.g. due to systematically high cloud
fractions). We transform this satellite-derived a priori tropospheric $NO_2$ vertical column
($V_{trop,prior}$) into slant column space using the tropospheric air mass factors ($A_{trop}$) provided with
the OMI data:
$$S_{trop,prior} = V_{trop,prior} \cdot A_{trop}$$     Equation 2.
Figure 1b shows our initial estimate of stratospheric vertical $NO_2$ columns over the TEMPO
domain resulting from the combination of Equation 1 and 2. We already see that this
stratospheric $NO_2$ estimate varies predominately as a function of latitude, although
anomalously low values are seen over some urban centers (e.g. around Los Angeles, Chicago,
and New York) where the a-priori tropospheric $NO_2$ slant column is large.
To exclude locations where this initial stratospheric vertical column estimate is likely
biased, we make use of the masking approach from Bucsela et al. (2013). This is based on
eliminating pixels where tropospheric contamination is high (or where the initial stratospheric
vertical column estimate would exceed the actual stratospheric vertical column by some
reasonable value) by requiring:
$$\frac{S_{trop,prior}}{A_{strat}} < 0.3 \times 10^{15} \ cm^{-2}$$     Equation 3.
On a typical day in July, this means that contamination from the troposphere would be less
than ~10% percent of the stratospheric $NO_2$ estimate (which generally ranges from 2-4 x $10^{15}$
$cm^{-2}$ over the TEMPO field of regard). Figure 1c shows the result of this masking step. The
threshold removes all the urban regions with anomalously low values in Figure 1b, in addition
to many other areas. Sensitivity tests show that the final stratospheric $NO_2$ estimate varies by
less than 5% for changes in this threshold between 0.2 x $10^{15}$ or 0.4 x $10^{15}$ $cm^{-2}$, consistent
with the generally small sensitivity found by Bucsela et al. (2013)). On this example day (and



for the month of July on average) the masking threshold of 0.3 x $10^{15}$ cm$^{-2}$ removes 55% of
the original data within the TEMPO field of regard. We find coverage is best over Canada and
over the Pacific Ocean, with less coverage over the rest of the continent and the Atlantic
Ocean. The original global algorithm removes ~28% of the available global data on average
for days in July, since tropospheric NO$_2$ columns are generally lower elsewhere in the world.

6       Since $S_{trop,prior}$ is calculated based on radiative transfer calculations ($A_{trop}$) in addition to

the a priori tropospheric NO$_2$ vertical column (Equation 2), this masking approach in principle
allows for polluted pixels to remain if the lower tropospheric signal is sufficiently suppressed
by clouds resulting in a low tropospheric air mass factor (or conversely excludes pixels with a
considerable tropospheric signal due to high surface reflectivity). We investigated the use of
explicitly cloudy scenes (cloud radiance fraction > 0.9), which could suppress the signal from
below. Mid-level clouds (600-400 hPa) are the least likely to contain significant NO$_x$ mixed
in from the surface, or lightning NO$_x$ associated with higher clouds. We find that most
(>75%) of the pixels that meet these criteria are already retained by our original masking
algorithm. Incorporating the remaining cloudy pixels to the masked data increases data
coverage by less than 1%. Given the uncertainties in retrieving cloud properties, uncertainties
in cloudy air mass factors, and the minimal added value of this dataset, we disregard adding
the remaining cloudy pixels to our algorithm.
In Bucsela et al. (2013), the remaining unmasked data are binned and un-filled bins are
interpolated using 2-dimensional averaging with a 30° longitude x 20° latitude moving
window. In our case, this step necessarily precludes information from outside the TEMPO
field of regard over the mostly pristine oceans from being used in the 2-D averaging. As we
will show, this leads to biases near the field of regard edges when compared to a global
algorithm, since the averaging window is disproportionately impacted by observations with
continental influence. We reduce this bias by incorporating independent global observations
from low Earth orbit that can provide context outside of the TEMPO field of regard. This
approach exploits the independent low Earth orbit observations that are expected throughout
the lifespan of TEMPO (e.g. GOME-2, TROPOMI).
Here, we employ GOME-2 observations as an independent dataset to estimate
stratospheric NO$_2$ at GOME-2 overpass time outside the TEMPO field of regard by using an
identical algorithm on this global data. We empirically transform the GOME-2 stratospheric
NO$_2$ estimate to the TEMPO observation time (here, the OMI overpass time), using the


climatological 30-day running mean local ratio of GOME-2 to OMI stratospheric $NO_2$. A
similar observational or model climatology could readily be constructed with TEMPO data
after launch based on the available low Earth orbit observations at the time. Figure 1d shows
the outcome of this approach. The GOME-2 observations outside of the TEMPO field of
regard retain the same magnitude and latitudinal gradient as the available observations within
the TEMPO field of regard, suggesting that the additional context from an independent low
Earth orbit instrument can be useful even when they are from a different time of day.
Before interpolating the unfilled bins, we apply a boxcar filter using a moving 15° x 10°
window as follows. First, our boxcar filter returns a smoothed array using the following
algorithm:

$$R_i = \frac{1}{w} \sum_{j=0}^{w-1} A_{i+j-w/2} \text{ where } \frac{(w-1)}{2} \le i \le N - \frac{(w+1)}{2}$$

11                                                         Equation 4

where $w$ is the smoothing width (in our case, defined in two dimensions by both a length and
width), $R_i$ is the $i$-th point in the smoothed data, and $A_i$ is the $i$-th point in the original data. For
data points where the neighborhood includes points outside the array, the nearest edge points
are used to compute the smoothed result. The variance of the original data is also calculated
using a similar algorithm. Any value that lies outside of the moving window average by ± 1.5
standard deviations is removed. While the Bucsela et al. (2013) algorithm uses the same
window size in a boxcar filtering step, it is performed later and only remove values above the
mean ("hotspots"). Here, we perform this boxcar filter in both directions (above and below
the mean) to remove anomalously low values that might result from a biased a-priori
tropospheric estimate that was not accounted for in the masking step (avoiding negative
stratospheric $NO_2$ values being retained in subsequent steps), and to remove anomalously
high values that might result from transient pollution events that were likewise missed in the
masking step. We perform this boxcar filter twice to strictly remove outliers from regions
with noisy data.

26       Missing bins are then interpolated using a 30° longitude x 20° latitude moving window.

We tested smaller window sizes and found that they could introduce unphysical variability,
and/or leave missing data. Figure 1e shows how all the missing data over the TEMPO domain
are successfully filled using this window size. A few remaining "hot spots" are accounted for
in a third pass of the boxcar filter.



To obtain our final stratospheric $NO_2$ column estimate, we apply a final simple
smoothing step with a 5° x 3° window, as in Bucsela et al. (2013). The smaller box-car
window size in this step recognizes, and allows for, some regional scale variability in the
stratosphere. Figure 1f shows the final stratospheric $NO_2$ column estimate over the TEMPO
field of regard. Variation is primarily a function of latitude, from around 2 x $10^{15}$ molec cm$^{-2}$
at the lowest latitudes in the field of regard (~20° latitude) to around 4 x $10^{15}$ molec cm$^{-2}$ at
the highest latitudes (~60° latitude). It is also apparent that this spatial filtering algorithm
allows for important regional scale variability to be retained in the stratospheric estimate.
Figure 2 shows the results of the same algorithm from an example day in January. The
shape of the expected TEMPO domain is impacted by large solar zenith angles at the highest
latitudes (we again use a solar zenith angle cut-off of 80°). Tropospheric enhancements
feature more prominently in the total slant column (Figure 2a) than in July since stratospheric
$NO_2$ columns are lower in the winter, and tropospheric $NO_2$ columns are higher. Figure 2b
shows the initial stratospheric estimate ($V_{init}$) from Equation 1, again using the monthly mean
GOME-2 tropospheric $NO_2$ column as an a priori estimate (Equation 2). Figure 2c shows the
result of applying the masking threshold (Equation 3). We find this threshold removes 51% of
the available data on average for this month (~21% of the available data are removed in the
global algorithm in January). Over the TEMPO domain we find that a slightly smaller fraction
pixels are removed in January compared to July because, despite having generally higher $NO_2$
tropospheric column densities, tropospheric air mass factors across the northeast are
extremely low at this time of year (discussed below). The low values are primarily due to
increased wintertime cloudiness. In this case, the masking threshold did not remove a strong
enhancement over the center of the continent. This highlights some criticism by Beirle et al.
(2016) of spatial filtering algorithms that rely strongly on a-priori climatologies wherein
transient tropospheric events could be misinterpreted as stratospheric. We find that varying
the magnitude of the threshold (Equation 3) does not successfully correct for this, since our
masking approach is based on a monthly mean and does not identify transient events, but this
feature is diminished in subsequent steps. Figure 2d shows the estimated stratospheric $NO_2$
outside of the TEMPO field of regard from the independent GOME-2 observations. Again,
these low Earth orbit observations provide powerful context despite being from a different
time of day. Figure 2e shows the result of the first two passes of the boxcar filter, and
interpolating unfilled bins using the 30° longitude x 20° latitude moving window.



Figure 2f shows the final stratospheric NO₂ estimate after the final pass of the statistical
test and 5° x 3° smoothing. The large enhancement of NO₂ over the continent has been
substantially dampened by our statistical filtering. The variability in the stratospheric NO₂
column is again generally latitudinal as expected, with values above $2 \times 10^{15}$ molec cm$^{-2}$ at
the low latitudes, and below $1 \times 10^{15}$ molec cm$^{-2}$ at the high latitudes.
The full TEMPO domain will have simultaneous sunlit coverage from about 1400 UTC
to 2300 UTC in July, and for only a few hours in January, based on a solar zenith angle
threshold of ~80°. Of concern is the lack of coverage over the west coast in the morning, and
over the east coast in the evening, where sunlit observations will not be available. Under these
circumstances, the stratospheric separation algorithm is challenged by even narrower spatial
domains. We evaluate these cases by repeating the calculations at specific times of day.
Figure 3 shows how the TEMPO algorithm would operate for 1130 Coordinated
Universal Time (UTC), 6:30 a.m. Eastern Standard Time (EST), on the example day in July.
Daylight observations over eastern North America are available by this time, without
coverage over the rest of the continent. All the algorithm steps are identical to those in Figure
1 and Figure 2 other than treatment of this partial coverage (additional near-real-time
considerations are discussed in Section 5). Figure 3a shows the OMI total slant columns. By
6:30 a.m. EST TEMPO observes only eastern North America. The availability of observations
increases in width northward because of the TEMPO viewing geometry. Figures 3b and 3c
show the initial stratospheric estimate (according to Equation 1) and the masked stratospheric
estimate (according to Equation 3) respectively. Figure 3d shows the independent low Earth
orbit observations from GOME-2 outside of the TEMPO field of regard. The observations are
binned, pass the statistical filtering steps, and interpolated in Figure 3e. The final stratospheric
estimate is shown in Figure 3f. Comparing this final stratospheric NO₂ estimate with the
estimate in Figure 1f (where coverage over the whole continent is assumed to be available),
we see the reduced coverage has negligible impact the final stratospheric estimate, and
identical spatial features are preserved ($R^2 = 0.995$).
Likewise, Figure 4 shows how the algorithm would operate on the example day in
January at 2330 UTC, or 3:30 pm Pacific Standard Time (PST). In addition to the loss of
observations in the east due to the time of day, larger solar zenith angles in the north at this
time of year further diminish coverage. Again, the subsequent steps are otherwise identical to
those in Figures 1 through 3. Figure 4a shows the OMI total slant columns. Observations are



available over parts of the Pacific Northwest, with coverage widening southward so that
observations are available from California to the western edge of Texas, and over western
parts of Mexico. Figure 4b and 4c show the initial stratospheric estimate (according to
Equation 1) and the masked stratospheric estimate (according to Equation 3) respectively.
Figure 4d shows how the independent low Earth orbit observations from again GOME-2
provide coverage outside of the TEMPO field of regard. After binning and interpolation
(Figure 4e) followed by hot spot removal and smoothing, the final TEMPO stratospheric
estimate is shown in Figure 4f. Comparing this stratospheric $NO_2$ estimate with Figure 2f
(where coverage over the whole continent is assumed to be available) demonstrates again how
the reduced coverage has negligible impact the final stratospheric estimate, and identical
spatial features are preserved ($R^2 = 0.997$).

12       Next, we examine in detail the potential information penalty associated with the limited

TEMPO field of regard compared to a global algorithm, and demonstrate quantitatively that
our approach can produce a tropospheric $NO_2$ estimate that is consistent with a global
algorithm, regardless of the time of day.

## 4    Stratosphere-Troposphere Separation over the TEMPO Field of Regard

18       The final step in the algorithm is the subtraction of the stratospheric $NO_2$ estimate from

the total slant column to obtain the tropospheric $NO_2$ column by:
$$V_{trop} = \frac{(S - V_{strat} \cdot A_{strat})}{A_{trop}} \qquad \text{Equation 5}$$
For this calculation we use the stratospheric and tropospheric air mass factors provided with
the OMI data product (the operational TEMPO algorithm would use TEMPO air mass
factors).

24       The difference between two tropospheric $NO_2$ column retrievals ($V_{trop,2}$ and $V_{trop,1}$) that

result from two different stratospheric $NO_2$ estimates ($V_{strat,2}$ and $V_{strat,1}$), but identical slant
columns and air mass factors, is directly proportional to the ratio of the tropospheric to
stratospheric air mass factors:
$$V_{trop,2} - V_{trop,1} = \frac{A_{strat}}{A_{trop}} \left( V_{strat,2} - V_{strat,1} \right) \quad \text{Equation 6}$$



This means that differences (or errors) in stratospheric $NO_2$ estimates are magnified in the
tropospheric $NO_2$ column depending on the local air mass factors. This issue is particularly
important over the eastern US in the winter, where tropospheric air mass factors can be very
low (<0.1), and stratospheric air mass factors can be high (~5) depending on viewing
geometry. Figure 5 shows the stratospheric and tropospheric air mass factors for January 15,
2007. Over areas of the eastern US, where clouds prevail, the tropospheric air mass factors are
exceedingly small (~0.01), which gives rise to extremely large $A_{strat}/A_{trop}$ ratios (>200). In
other words, residuals between two stratospheric $NO_2$ algorithms can become magnified by
more than two orders of magnitude in the troposphere.
The impact of errors in the tropospheric column due this issue can be minimized by
excluding observations with high stratospheric to tropospheric air mass factor ratios. This is
also based on the logic that such values indicate tropospheric $NO_2$ is making a small
contribution to the measured signal (and as a result, the tropospheric $NO_2$ retrieval should
have high uncertainty). For this reason, we restrict all tropospheric $NO_2$ estimates to where
the local stratospheric to tropospheric air mass factor ratios are less than 5.
Figure 6 shows the stratospheric and tropospheric $NO_2$ columns estimated for July 15,
2007. The top panels display the stratospheric and tropospheric $NO_2$ columns as derived from
our TEMPO algorithm that employs the OMI data as a surrogate for TEMPO observations,
with adjacent GOME-2 data provided context outside the field of regard. The middle panels
display the stratospheric and tropospheric columns derived from implementing our algorithm
globally with OMI data alone (the results are restricted to the TEMPO field of regard in the
figure to facilitate comparison). The bottom panel shows the differences between our TEMPO
algorithm and the global algorithm. We find excellent spatial agreement in the tropospheric
$NO_2$ estimate between the two algorithms ($R^2$ = 0.997, slope = 1.008). More than 95% of the
pixels have differences that are smaller than ± 0.1 x $10^{15}$ molec $cm^{-2}$.
Figure 7 compares the stratospheric and tropospheric $NO_2$ column estimates from the
TEMPO and global algorithms for January 15, 2007. The loss of coverage in the troposphere
(mostly over the eastern US) is a result of the air mass factor issue discussed above, leading to
tropospheric $NO_2$ retrievals with low information content. The spatial agreement in the
tropospheric $NO_2$ estimates that remain is excellent across the domain ($R^2$ = 0.996 slope =
0.999). The magnitude of the differences in the stratospheric columns become larger in the





troposphere, exceeding 0.5 x $10^{15}$ molec cm$^{-2}$ near the edges. Nonetheless, ~95% of the pixels
are consistent with the global version of the algorithm to within 0.25 x $10^{15}$ molec cm$^{-2}$.

3        Figure 8 shows the monthly mean tropospheric NO$_2$ columns resulting from our TEMPO

stratosphere-troposphere separation algorithm for both July and January, and the difference
versus results from the global algorithm. We find that our TEMPO algorithm produces
monthly mean results with negligible difference compared to the global algorithm, even at the
field of regard edges. The correlation between the two algorithms is excellent ($R^2$ = 0.999 and
slope = 1.009 for July, $R^2$ = 0.998 and slope = 0.999 for January). For July, more than 99% of
the pixels have differences that are smaller than ±0.05 x $10^{15}$ molec cm$^{-2}$. For January, more
than 90% of the pixels have differences that are smaller than ±0.05 x $10^{15}$ molec cm$^{-2}$, and
more than 99% of the pixels have differences that are smaller than ±0.10 x $10^{15}$ molec cm$^{-2}$. In
other words, our TEMPO-specific algorithm performs almost identically to the low Earth
orbit algorithm that uses all available global data. There are some random errors near the field
of regard edges on individual days (Figures 6 and 7), but these nearly disappear in the
monthly average (Figure 8)

16        Figure 9 shows the July monthly mean tropospheric NO$_2$ columns resulting from

retrievals at 1130 UTC (east coast summer morning) and at 0200 UTC (west coast summer
evening). The east coast morning retrieval example exhibits small positive biases over some
the Great Lakes region compared to the global algorithm, but overall the spatial agreement
remains excellent ($R^2$ = 0.996 and slope = 1.015). More than 90% of the pixels have
differences that are smaller than ±0.05 x $10^{15}$ molec cm$^{-2}$, and more than 98% of the pixels
have differences that are smaller than ±0.10 x $10^{15}$ molec cm$^{-2}$. The west coast summer
evening example also exhibits excellent performance overall ($R^2$ = 0.998 and slope = 0.994).
In this case, more than 98% of the pixels have differences that are smaller than ±0.05 x $10^{15}$
molec cm$^{-2}$.

26        Figure 10 shows the January monthly mean tropospheric NO$_2$ columns resulting from

retrievals at 1400 UTC (east coast winter morning) and 2330 UTC (west coast winter
evening). The bottom panels in Figure 10 show the difference between the results from our
TEMPO algorithm and the results from the global algorithm. In the east coast winter case,
spatial agreement is still very good in general ($R^2$ = 0.995), but we find noticeable
degradation in the absolute performance over the continent compared to the global algorithm
resulting from this partial field of view (slope = 1.038). The west coast winter evening





retrieval performs better overall ($R^2$ =0.999, slope = 1.007). Although the algorithm performs
poorest in the east coast winter morning case, ~90% of the tropospheric pixels still have
differences that are less than $0.2 \times 10^{15}$ molec $cm^{-2}$, a commonly accepted estimate of the
stratospheric uncertainty resulting from stratosphere-troposphere separation in $NO_2$ retrieval
algorithms (Boersma et al. 2004). Moreover, two hours later at 1600 UTC when the field of
regard has expanded across the Great Lakes region, into the middle of North America, and
covers most of Mexico, this issue disappears ($R^2$ = 0.999, slope = 0.998). In other words, as
spatial coverage expands, the absolute constraint on stratospheric $NO_2$ becomes more robust.
This highlights the challenge of accurate wintertime tropospheric $NO_2$ retrievals
(especially over eastern North America) when pollution is primarily in a shallow boundary
layer close to the surface where satellite remote sensing sensitivity is lowest. The partial
TEMPO field of regard in this case exacerbates the problem, but the challenge is not unique
to TEMPO retrievals.
Finally, we further test the performance of this algorithm at other times of day by
repeating the same steps as above, but using GOME-2 observations as a surrogate for
TEMPO. For this, we swap all instances of the OMI observations (overpass time ~ 13:30)
with GOME-2 observations (overpass time ~09:30), and vice versa. In other words, the
GOME-2 observations are restricted to the anticipated field of regard, and we use a monthly
from OMI as our a priori tropospheric column and the daily observations from OMI as
supporting global observations outside the TEMPO field of regard. We find the performance
at this morning overpass time is as good as the mid-afternoon overpass time ($R^2$ = 0.999,
slope = 1.005 for July; and $R^2$ = 0.999, slope = 1.005 for January), providing more evidence
that our approach works equally well at different times of day.
**5    Near-Real-Time Considerations**
For retrievals in near-real time (i.e. within an hour of the observation), independent
global observations in low Earth orbit may not be available (e.g. unexpected issues with low
Earth orbit observation processing). Here we test the performance of the TEMPO algorithm
without the supporting global observations by carrying out the identical steps outlined in
Sections 3 and 4 except without incorporating the GOME-2 observations outside the TEMPO
field of regard. Comparing these results with the global algorithm isolates the penalty due to





the limited TEMPO spatial domain alone, since the steps are otherwise computationally
identical.

3        Figure 11 shows the mean July and January tropospheric columns resulting from this
near-real time test. The spatial correlation with the global algorithm is still strong overall ($R^2$
= 0.924 and slope = 0.973 for July and $R^2$ = 0.996 and slope = 1.008 for January), and
between 90-95% of pixels in both July and January differ from the global algorithm by less
than $0.2 \times 10^{15}$ molec cm$^{-2}$. We find that, compared to a global algorithm, this stratosphere-
troposphere separation approach gives rise to noticeable systematic biases near the field of
regard edges (including Mexico, the Caribbean, and northern Canada). The differences are
due to the lack of supporting data outside of the TEMPO field of regard.

11       This is most evidently a problem near the northern/southern borders of the field of regard,
given the strong gradient in stratospheric $NO_2$ as a function of latitude. At low latitudes, when
the averaging windows intersect with the field of regard, the global algorithm would have
lower mean values by including observations to the south. This causes the stratospheric
column from the TEMPO algorithm to be systematically biased high compared to the global
algorithm, translating into an underestimate in the tropospheric column (by more than $-0.5 \times$
$10^{15}$ molec cm$^{-2}$ in some locations). By the same logic, there is a high bias (also more than
$+0.5 \times 10^{15}$ molec cm$^{-2}$ on average) along the northern edge of the field of regard in July.
There are also small low biases in the tropospheric column throughout the eastern side of the
TEMPO field of regard over the Atlantic Ocean. By excluding more pristine ocean conditions
further to the east, the stratospheric column derived by the TEMPO algorithm is biased high
compared to the global algorithm, which again translates into an underestimate in the
tropospheric column.

24       In the absence of daily ancillary satellite data for estimating stratospheric $NO_2$ outside the
field of regard, a climatology built from satellite observations or model data could mitigate
these edge effects for near real time retrievals since the average latitudinal and seasonal
dependence of stratospheric $NO_2$ are generally well known. For example, tests conducted
using a monthly mean global stratospheric $NO_2$ estimate as the supporting data outside the
TEMPO field of regard improves the correlations in both cases ($R^2$ = 0.999 and slope = 1.010
for July and $R^2$ = 0.999 and slope = 1.002 for January), now with >99% of the monthly mean
pixels differing from the global algorithm results by less than $0.05 \times 10^{15}$ molec cm$^{-2}$.





Similarly, we find weaker overall performance in the cases of partial fields of regard
without context from surrounding low Earth orbit observations. Figure 12 shows the July
mean tropospheric column retrievals calculated for 1130 UTC (east coast summer morning)
and the July mean tropospheric column retrievals for 0200 UTC (west coast summer
evening). Though this version of the algorithm performs less well compared to the results
from incorporating independent low Earth orbit observations, the spatial correlation is still
good ($R^2$ = 0.944, slope = 0.943 for 1130 UTC July; $R^2$ = 0.964, slope = 0.986 for 0200
UTC). The differences over most of the available domain remain small, with 90-95% of the
pixels having differences in the mean tropospheric column of less than ± 0.2 x $10^{15}$ molec cm$^-$
$^2$ compared to the global algorithm. Figure 13 shows the January mean tropospheric column
retrievals calculated for 1400 UTC (east coast winter morning) and the January mean
tropospheric column retrievals for 2300 UTC (west coast winter evening). The spatial
correlation in both cases remains strong, again with some systematic biases observed ($R^2$ =
0.996, slope = 1.001 at 1400 UTC and $R^2$ = 0.987, slope = 1.019 at 2330 UTC). The biases
remain modest, with ~90% of the pixels being consistent to within 0.2 x $10^{15}$ cm$^{-2}$ of the
global implementation of the algorithm. Again, using a monthly climatology mitigates the
biases in all cases, with the smallest improvement for the retrieval in January at 1400 UTC
(going from 90% to 94% of the pixels being consistent to within 0.2 x $10^{15}$ cm$^{-2}$ of the global
implementation of the algorithm).
**6   Conclusions**
The TEMPO geostationary satellite instrument is expected to provide hourly observations
of $NO_2$ columns (among a variety of other measurements) over North America. Here, we have
developed and tested the first stratosphere-troposphere separation algorithm for TEMPO
geostationary satellite observations of atmospheric $NO_2$ column density. We use independent
measurements from a low Earth observing satellite instrument to identify likely locations of
tropospheric enhancements, and to provide context outside of the available TEMPO
measurements. We consider partial fields of regard as a function of time of day, and
implement a new filter based on stratospheric to tropospheric air mass factor ratios. We
investigate in particular the information penalty associated with the limited TEMPO fields of
regard as a function of season and time of day.



We find that our algorithm performs as well as a global low Earth orbit algorithm for
most scenarios. When the whole continent is observed, monthly mean agreement with
tropospheric $NO_2$ retrieved from the global algorithm is excellent ($R^2 = 0.999$, slope = 1.009
for July and $R^2 = 0.998$, slope = 0.999 January). During most instances with a partial field of
regard (e.g. east coast morning or west coast evening) the algorithm still performs robustly.
We demonstrate that small biases near the southern and northern edges of the field of regard
are avoided by incorporating independent low Earth orbit observations that have been
corrected for the time of day. When the whole continent is observed, the vast majority of
pixels (> 95%) agree with results from a global implementation of the same algorithm to
within $\pm\ 0.05 \times 10^{15}$ molecules cm$^{-2}$. We find that the TEMPO algorithm is challenged most
by winter east coast morning retrievals, but nonetheless the difference between the TEMPO
algorithm and the global implementation of the same algorithm produces differences that are
less than $0.2 \times 10^{15}$ molecules cm$^{-2}$ for more than 90% of the pixels. Even when supporting
observations from low Earth orbit may not be available (as in near-real-time), a large majority
of pixels (~90% or greater) agree with the global algorithm to within $\pm\ 0.2 \times 10^{15}$ molecules
cm$^{-2}$ on a monthly mean basis, which is generally accepted as typical estimates of
stratospheric error due to stratosphere-troposphere separation algorithms. The differences can
be reduced further in near-real-time retrievals by the use of a climatology outside the TEMPO
field of regard. The value of independent low Earth orbit observations for TEMPO
tropospheric retrievals implies benefit to TEMPO data from ongoing development of low
Earth orbit observations.
We have demonstrated a feasible and robust stratosphere-troposphere separation
algorithm for the retrieval of geostationary satellite-based $NO_2$ tropospheric column densities
by the TEMPO instrument notwithstanding the limited field of regard or changing time of
day. This approach may be applicable to other planned geostationary satellite instruments
including Sentinel-4 over Europe and GEMS over Asia. This spatial filtering and
interpolation method may also have applications in offset removal during retrievals of HCHO
and $SO_2$ tropospheric columns.
**Acknowledgements**
The authors are grateful to Kelly Chance, Xiong Liu, John Houck, Peter Zoogman, other
members of the TEMPO trace gas retrieval team for their input in preparation of this



manuscript. Work at Dalhousie University was supported by Environment and Climate
Change Canada. The authors also gratefully acknowledge the free use of TEMIS $NO_2$ data
from the GOME-2 sensor provided by www.temis.nl, and the NASA Standard Product $NO_2$
data from OMI provided by http://disc.sci.gsfc.nasa.gov/Aura/data-
holdings/OMI/omno2_v003.shtml.

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





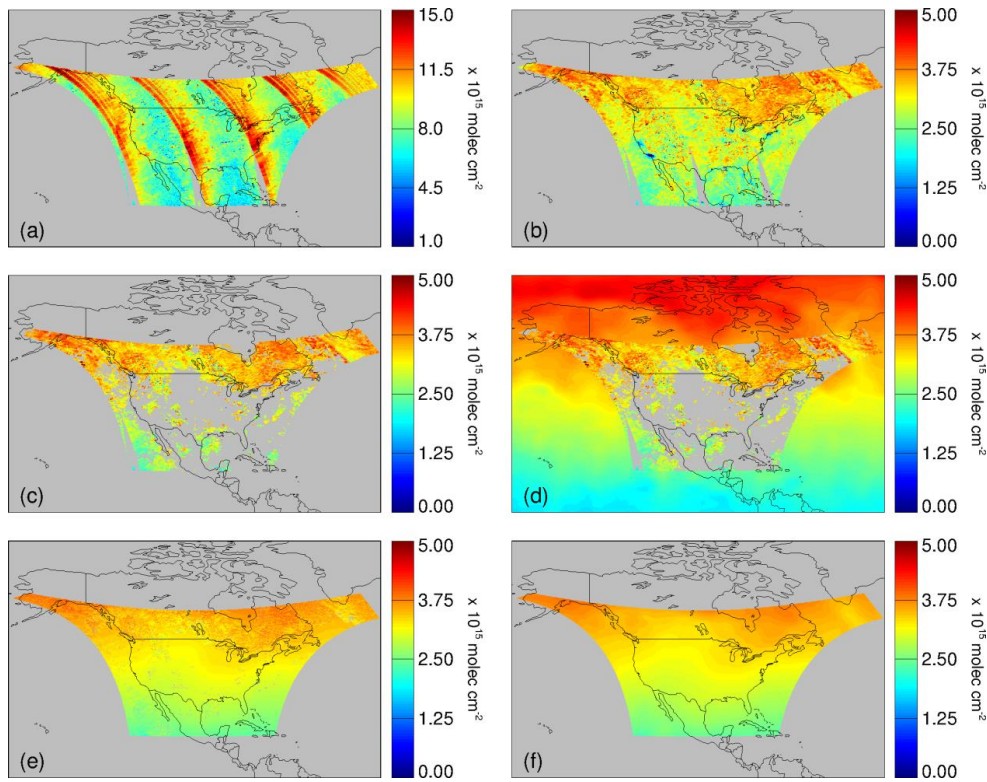

**Figure 1:** Calculation of the stratospheric NO$_2$ estimate on July 15, 2007 using OMI observations from within the anticipated TEMPO field of regard: (a) Slant columns on a 0.1° x 0.1° grid. (b) Initial stratospheric estimate ($V_{init}$) resulting from Equation 1 and 2. (c) Masked $V_{init}$ using a threshold of $S_{trop}/A_{strat} < 0.3$ x $10^{15}$ molec cm$^{-2}$ to remove large tropospheric influence. (d) Adding context outside of the TEMPO field of regard by using independent low-earth orbit observations from GOME-2 that have been corrected for time of day. (e) Stratospheric NO$_2$ estimate with masked areas interpolated. (f) Stratospheric NO$_2$ estimate after final hot spot removal and smoothing.

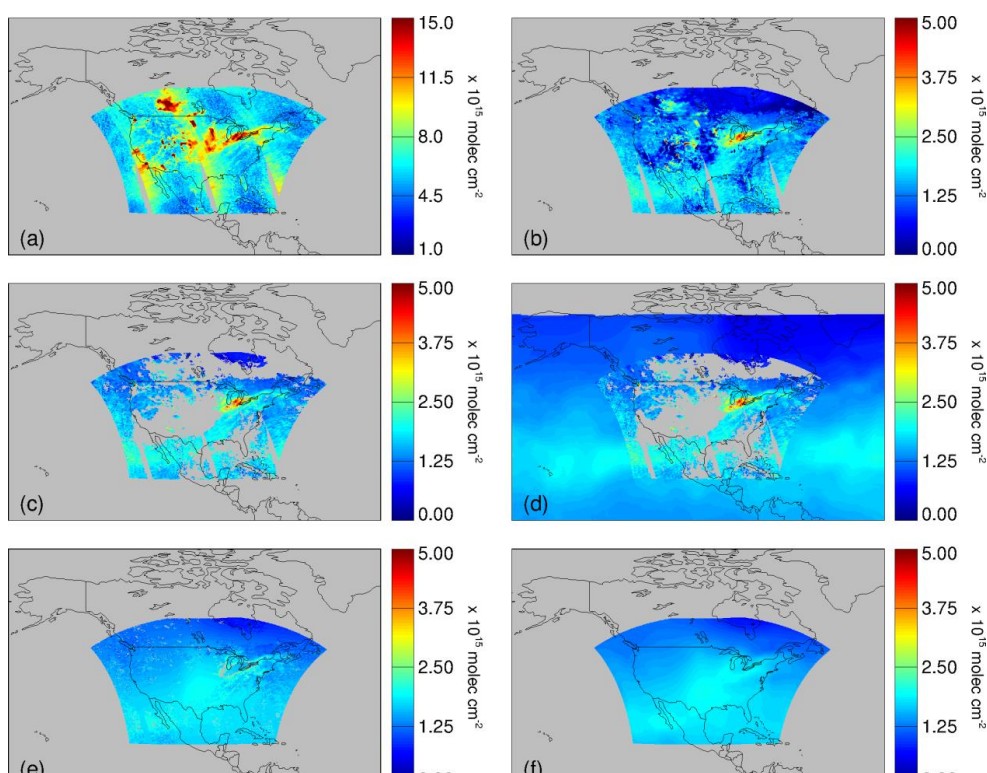

**Figure 2:** Calculation of the stratospheric $NO_2$ estimate on January 15, 2007 using OMI observations from within the anticipated TEMPO field of regard: (a) Slant columns on a 0.1° x 0.1° grid. (b) Initial stratospheric estimate ($V_{init}$) resulting from Equation 1 and 2. (c) Masked $V_{init}$ using a threshold of $S_{trop}/A_{strat} < 0.3$ x $10^{15}$ molec cm$^{-2}$ to remove large tropospheric influence. (d) Adding context outside of the TEMPO field of regard by using independent low-earth orbit observations from GOME-2 that have been corrected for time of day. (e) Stratospheric $NO_2$ estimate with masked areas interpolated. (f) Stratospheric $NO_2$ estimate after final hot spot removal and smoothing.



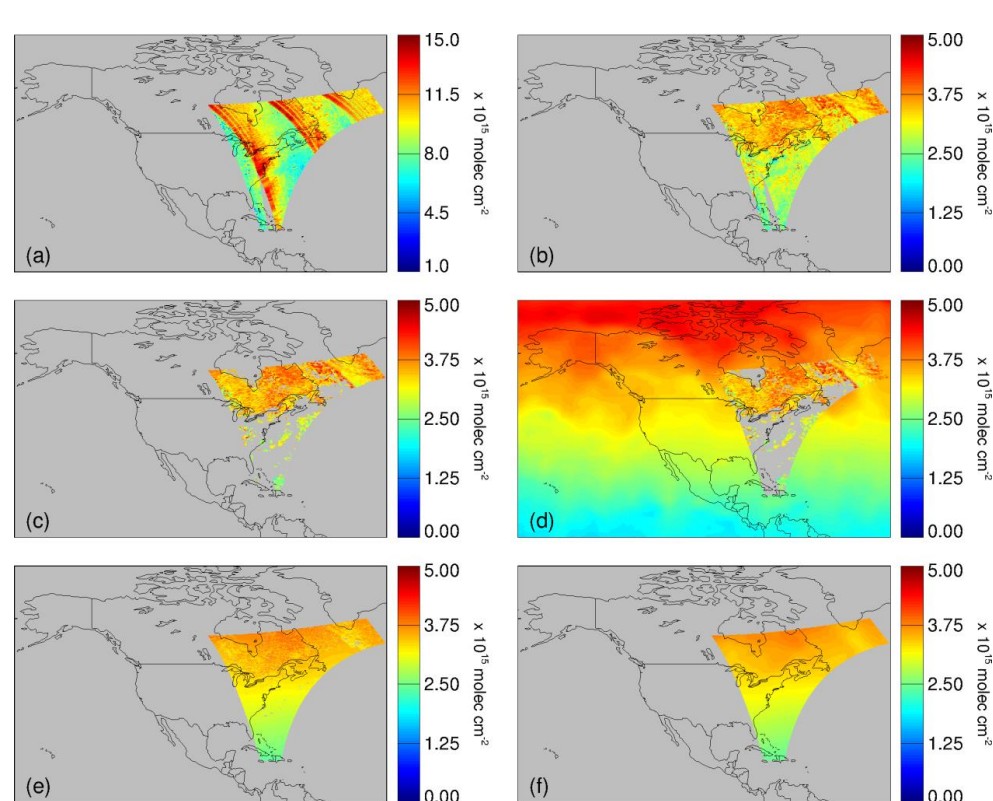

**Figure 3:** Calculation of the stratospheric NO$_2$ estimate on July 15, 2007 using OMI

observations from within the anticipated TEMPO field of regard at 1130 UTC (6:30 am

Eastern Standard Time): (a) Slant columns on a 0.1° x 0.1° grid. (b) Initial stratospheric

estimate ($V_{init}$) resulting from Equation 1 and 2. (c) Masked $V_{init}$ using a threshold of $S_{trop}/A_{strat}$

< 0.3 x 10$^{15}$ molec cm$^{-2}$ to remove large tropospheric influence. (d) Adding context outside of

the TEMPO field of regard by using independent low-earth orbit observations from GOME-2

that have been corrected for time of day. (e) Stratospheric NO$_2$ estimate with masked areas

interpolated and smoothed. (f) Stratospheric NO$_2$ estimate after final hot spot removal

smoothing.





**Figure 4:** Calculation of the stratospheric $NO_2$ estimate on January 15, 2007 using OMI

observations from within the anticipated TEMPO field of regard at 2330 UTC (3:30 pm

Pacific Standard Time): (a) Slant columns at 0.1° x 0.1° resolution. (b) Initial stratospheric

estimate ($V_{init}$) resulting from Equation 2. (c) Masked $V_{init}$ using a threshold of $S_{trop}/A_{strat} < 0.3$

x $10^{15}$ molec cm$^{-2}$ to remove large tropospheric influence. (d) Adding context outside of the

available TEMPO field of regard by using independent low-earth orbit observations from

GOME-2 that have been corrected for time of day. (e) Stratospheric $NO_2$ estimate with

masked areas interpolated and smoothed. (f) Final stratospheric $NO_2$ estimate after hot spot

removal and smoothing.



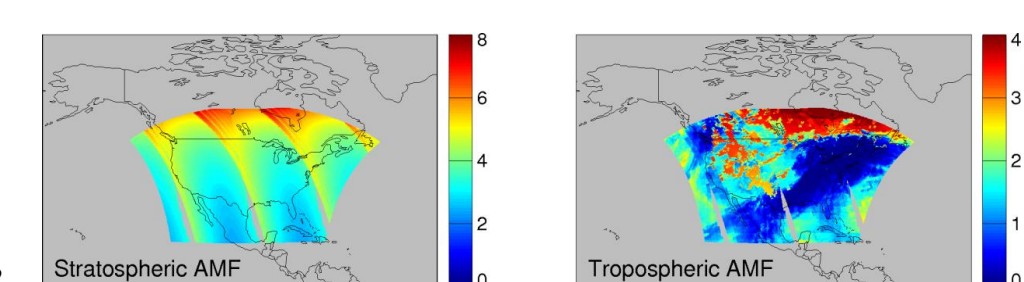

3 **Figure 5**: Stratospheric (left) and tropospheric (right) air mass factors for January 15, 2007.



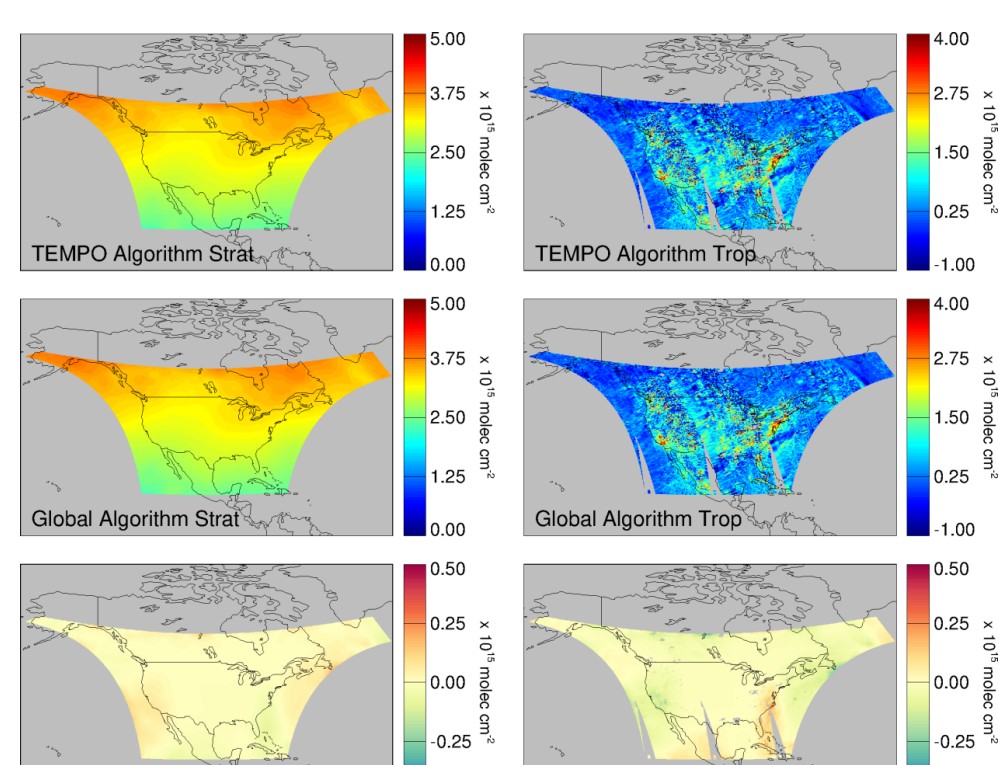

**Figure 6:** Stratospheric $NO_2$ (left panels) and final tropospheric $NO_2$ retrievals (right panels) resulting from our stratosphere-troposphere separation algorithms for July 15, 2007. Top panels show the results using our proposed TEMPO algorithm. Middle panels show the results using global observations (results have been clipped to the TEMPO field of regard for comparison). Bottom panels show the absolute absolute differences between the TEMPO and global algorithm results.



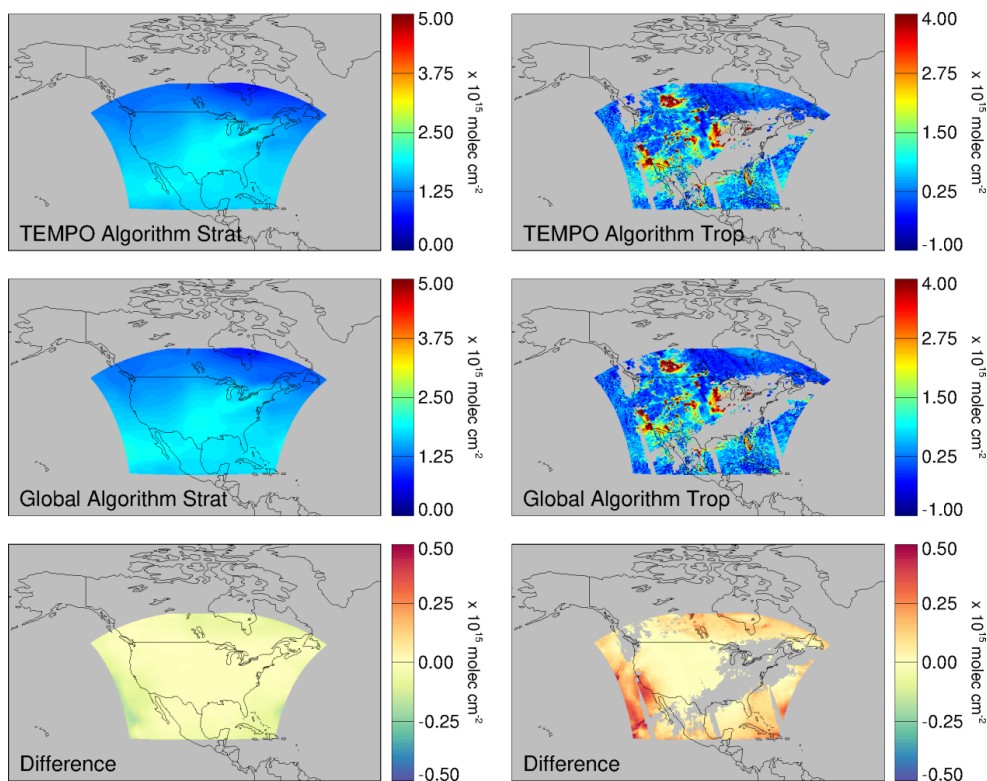

**Figure 7:** Stratospheric $NO_2$ (left panels) and final tropospheric $NO_2$ retrievals (right panels)

resulting from our algorithm for January 15, 2007. Top panels show the results using our

proposed TEMPO algorithm. Middle panels show the results using global observations

(results have been clipped to the TEMPO field of regard for comparison). Bottom panels

show the absolute differences between the TEMPO and global algorithm results.



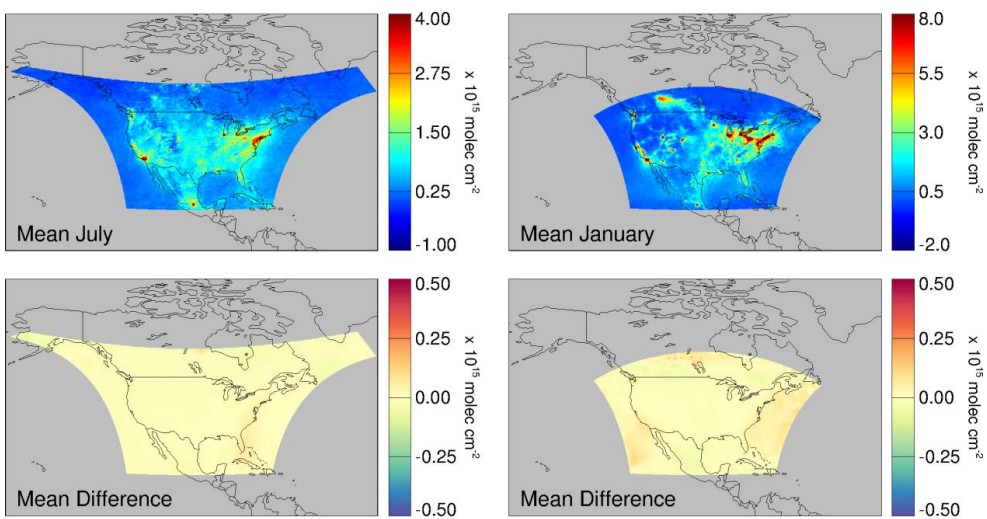

**Figure 8:** Top panels show mean July and January tropospheric $NO_2$ column densities resulting from our TEMPO algorithm. Bottom panels show absolute difference in mean July and January tropospheric $NO_2$ between the TEMPO algorithm and the global algorithm.



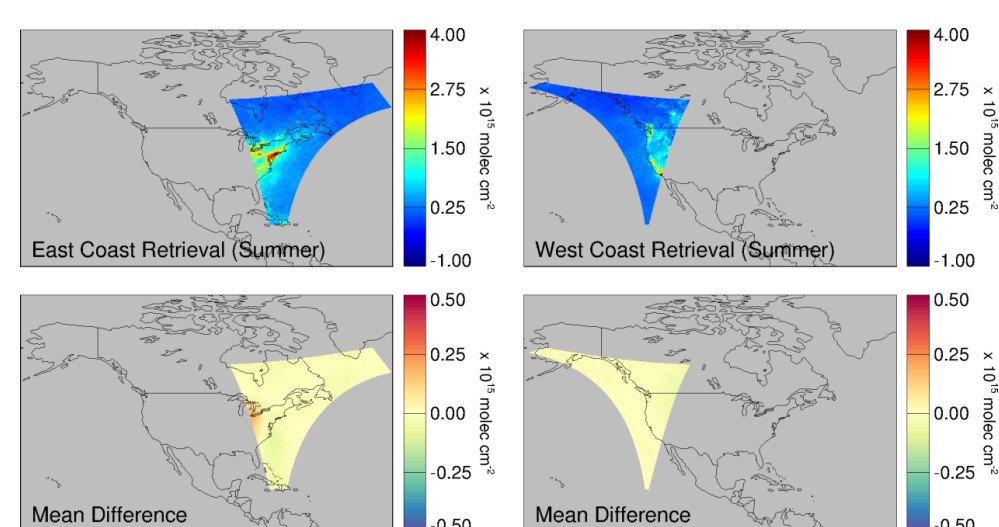

**Figure 9:** Top panels show mean July tropospheric $NO_2$ column densities at 1130 UTC (left)
and 0200 UTC (right) resulting from our TEMPO STS algorithm. Bottom panels show
absolute difference in the tropospheric $NO_2$ column between the TEMPO algorithm and the
global STS algorithm.



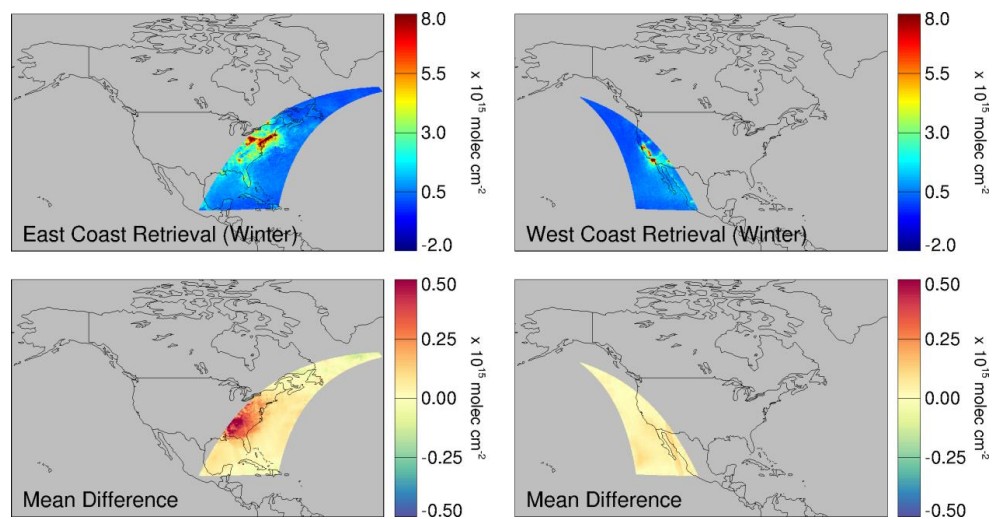

**Figure 10:** Top panels show mean January tropospheric $NO_2$ column densities at 1400 UTC
(left) and 2330 UTC (right) resulting from our TEMPO STS algorithm. Middle panels show
absolute difference in the tropospheric $NO_2$ column between the TEMPO algorithm and the
global STS algorithm.



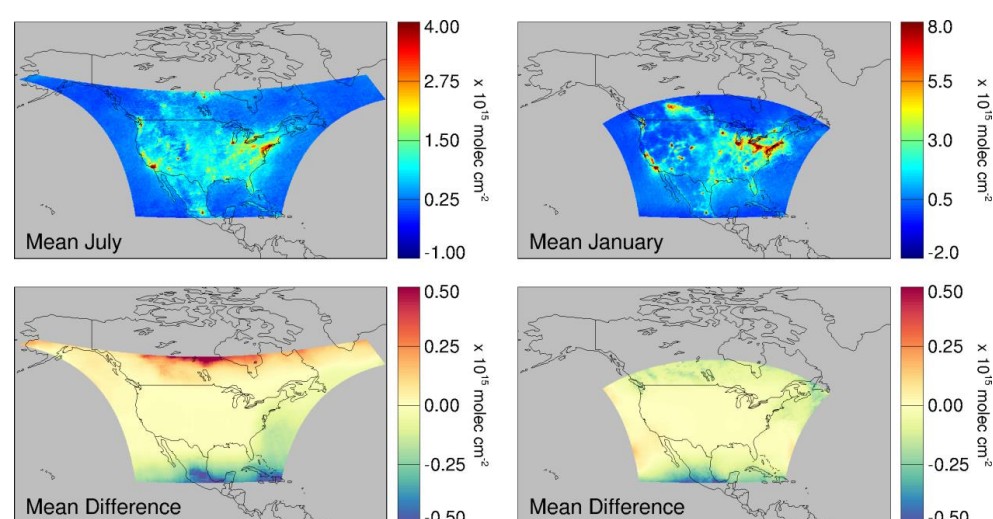

**Figure 11:** Top panels show mean July and January tropospheric $NO_2$ column densities resulting from our TEMPO STS algorithm without using independent low-earth orbit observations for context outside the TEMPO field of regard (as might be occasionally expected in near-real-time operations). Bottom panels show absolute difference in mean July and January tropospheric $NO_2$ between the TEMPO algorithm and the global STS algorithm.



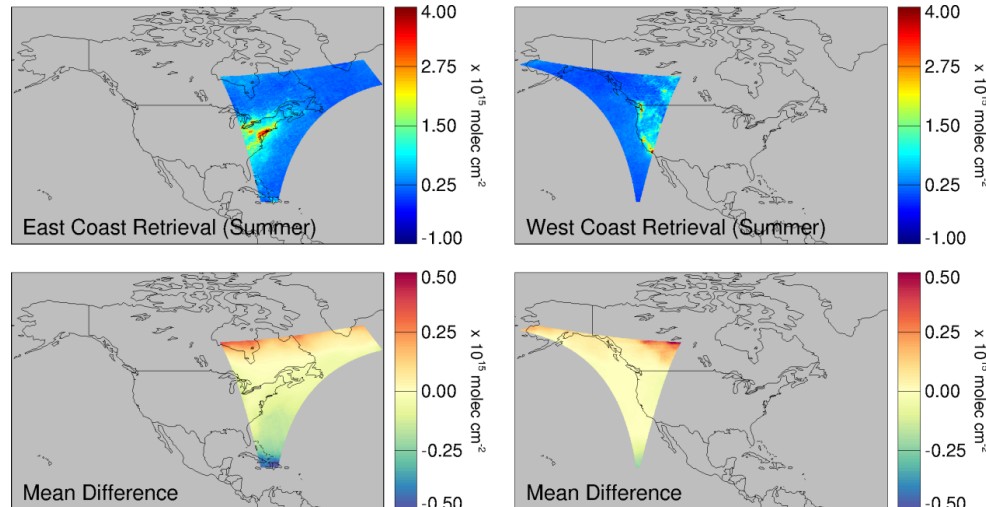

**Figure 12:** Top panels show mean July tropospheric $NO_2$ column densities at 1130 UTC (left) and 0200 UTC (right) resulting from our TEMPO STS algorithm without using independent low-earth orbit observations for context outside the TEMPO field of regard. Bottom panels show absolute difference in the tropospheric $NO_2$ column between the TEMPO algorithm and the global STS algorithm.



**Figure 13:** Top panels show mean January tropospheric $NO_2$ column densities at 1400 UTC

(left) and 2330 UTC (right) resulting from our TEMPO algorithm without using independent

low-earth orbit observations for context outside the TEMPO field of regard. Middle panels

show absolute difference in the tropospheric $NO_2$ column between the TEMPO algorithm and

the global algorithm.

