# Peer review of "Stratosphere-troposphere separation of nitrogen dioxide"

_Atmospheric Measurement Techniques, 2018_

## Referee Comment (RC1) · Anonymous Referee #1 · 9 Aug 2018

This paper presents a standard stratosphere-troposphere separation algorithm for the observations of NO2 from the TEMPO (Tropospheric Emissions: Monitoring of Pollution) satellite instrument. TEMPO, which will be launched between 2019-2021, will provide space-based measurements in geostationary orbit with a field of regard over North America from southern Canada to Mexico City and the Bahamas. Algorithm developments include the use of independent satellite observations (OMI and GOME-2) for identifying likely locations of tropospheric enhancements and for spatial context, the consideration of diurnally varying partial fields of regard, and a filter based on stratospheric to tropospheric air mass factor ratios. This algorithm is tested with Low Earth Orbit (LEO) from the OMI and GOME-2 satellite instruments. The potential information

penalty associated with the limited TEMPO field of regard compared to an identical global algorithm is also examined.

This study fits well with the scope of AMT and the manuscript is well written and clearly structured. Figures are also of very good quality. I recommend publishing the paper in AMT after addressing the following comments:

General concerns:

1/In the absence of daily independent satellite observations for the near-real-time processing, the back-up solution will be to use a climatology built on satellite observations or model data. Then, what will be the level of homogeneity/consistency of the retrieved TEMPO NO2 column time-series since they will consist in a combination of retrievals performed using different sources of ancillary data ? Do you foresee an offline reprocessing based on a unique source of ancillary data ? Or this is something which is not needed since this effect will be within the typical stratospheric error due to stratosphere-troposphere separation methods ?

2/The validation of the separation algorithm is not discussed at all in the paper. I think that at a later stage, it will be useful to compare the stratospheric NO2 column estimates with independent reference measurements, e.g. from ground-based DOAS UV-visible spectrometers. As first verification, maybe it would be interesting to compare within the anticipated TEMPO field of regard the estimates of the stratospheric NO2 vertical column with those included in the OMI and GOME-2 data products used in this study.

Both points 1/ and 2/ should be further discussed in the paper.

Specific comments:

Page 6, line 5: a short justification is needed about the fact that data are restricted to SZA smaller than 80° .

Page 7, line 1-4: Monthly mean of GOME-2 tropospheric NO2 columns is used as

initial a-priori tropospheric NO2 estimate. How is it done in practice ? Are the GOME-2 data first gridded on the same 0.1°x0.1° regular grid as OMI ? A clarification would be helpful here or at the end of the description of the GOME-2 data in Section 2. Also, since the tropospheric NO2 column can show strong diurnal changes, is the GOME-2 tropospheric column a good estimate of the column at the OMI overpass time ?

Technical corrections:

Page 3, line 6: 'Richter et al., 2005' instead of 'Richter et al. 2005'. Similar corrections should be done on the same page at lines 7, 13, 14; on page 3, line 3; on page 4, line 20; on page 6, line 2.

Page 4, line 19: 'available' instead of 'avialable'

Page 7, line 30: one bracket should be removed after '2013'.

---

## Referee Comment (RC2) · Anonymous Referee #2 · 25 Sep 2018

The paper describes the adaptation of a stratosphere/troposphere separation algorithm to the upcoming geostationary satellite instrument TEMPO. It is well written, logically structured and convincing in its conclusions. The paper should be published on AMT after dealing with the following issues:

General comments:

1. Gridding approach

The authors perform a gridding as very first step (page 5, line 29). This is not optimal, as satellite pixels with potentially very different conditions (i.e. a low total column over a clouded pixel next to a high total column over a power plant stack without clouds, both

within the same 0.1° grid box) are just averaged, with consequences hard to foresee due to the many nonlinearities involved. I would like to encourage the authors to rethink this approach and go for a different order, i.e. applying the filter on Strop,prior and the masking of pixels high ratio of strat vs trop AMF on individual satellite pixels rather than averaged 0.1° grid pixels.

2. Tropospheric columns

Please provide some information on the frequency distribution of tropospheric columns over remote regions. Do negative columns occur? How large and how variable is the tropospheric column at the edges of the TEMPO domain?

Minor issues

- after introducing LEO on page 1, line 19, please use it (e.g. page 2, line 9; page 3, line 1).

- please comment which STS algorithm is foreseen for operational processing of TEMPO.

---

## Author Comment (AC1) · 17 Oct 2018

We thank the reviewer for their comments, and respond to each below.

**Reviewer Comment (RC):** This paper presents a standard stratosphere-troposphere separation algorithm for the observations of NO2 from the TEMPO (Tropospheric Emissions: Monitoring of Pollution) satellite instrument. TEMPO, which will be launched between 2019-2021, will provide space-based measurements in geostationary orbit with a field of regard over North America from southern Canada to Mexico City and the Bahamas. Algorithm developments include the use of independent satellite observations (OMI and GOME-2) for identifying likely locations of tropospheric enhancements and for spatial context, the consideration of diurnally varying partial fields of regard, and a filter based on stratospheric to tropospheric air mass factor ratios. This algorithm is tested with Low Earth Orbit (LEO) from the OMI and GOME-2 satellite instruments. The potential information penalty associated with the limited TEMPO field of regard compared to an identical global algorithm is also examined.

This study fits well with the scope of AMT and the manuscript is well written and clearly structured. Figures are also of very good quality. I recommend publishing the paper in AMT after addressing the following comments.

**Author Response (AR):** We thank the reviewer very much for their positive and constructive remarks.

**RC:** In the absence of daily independent satellite observations for the near-real-time processing, the back-up solution will be to use a climatology built on satellite observations or model data. Then, what will be the level of homogeneity/consistency of the retrieved TEMPO NO2 column time-series since they will consist in a combination of retrievals performed using different sources of ancillary data? Do you foresee an offline reprocessing based on a unique source of ancillary data? Or this is something which is not needed since this effect will be within the typical stratospheric error due to stratosphere-troposphere separation methods?

**AR:** We thank the reviewer for the opportunity to clarify our strategy. Although we expect any effect near the field of regard edges to be small (as the reviewer points out), we do recommend an offline (or even night time) re-processing of the data using a unique source of ancillary data for outside the field of regard to avoid any inconsistencies in the retrieval over time. Meanwhile, as we demonstrate in the manuscript, a retrieval using the 30-day climatology will produce satisfactory results for near-real-time products.

In response to this reviewer comment, we have added the following text to our manuscript:

Page 18, Line 9:

*"Given these results, our recommendation for TEMPO is to use a climatological estimate (e.g. a 30-day mean) of stratospheric NO2 for context outside of the TEMPO field of regard during near-real-time retrieval if LEO observations are unavailable. This climatological estimate can be constructed based on satellite-derived observations in LEO from the preceding year and corrected for the time of day based on model results or other independent observations. We would then propose a subsequent re-processing of the data that incorporates the daily LEO observations when available from the correct observation day."*

**RC:** The validation of the separation algorithm is not discussed at all in the paper. I think that at a later stage, it will be useful to compare the stratospheric NO2 column estimates with independent reference measurements, e.g. from ground-based DOAS UV-visible spectrometers. As first verification, maybe it would be interesting to compare within the anticipated TEMPO field of regard the estimates of the stratospheric NO2 vertical column with those included in the OMI and GOME-2 data products used in this study.

**AR:** We agree with the reviewer that validation of the algorithm with independent reference measurements, including ground-based DOAS UV-vis spectrometers, will be useful to pursue. As the reviewer suggests, an initial option for now would be to compare the TEMPO stratospheric estimate with the stratospheric NO2 estimates already calculated by OMI and GOME-2 algorithms.

In response to this reviewer comment, we have performed this initial evaluation, and added the following text to the manuscript:

Page 10, Line 18:

*"In an effort to evaluate our new TEMPO algorithm with an independent estimate, we compare our stratospheric vertical column with the stratospheric vertical column included in the OMI SPv3 retrieval. Despite using different prior tropospheric estimates, incorporating observations from GOME-2 outside the field of regard during interpolation, and employing different box-car filtering steps, our algorithm is highly consistent with the results from the global NASA standard OMI product over the TEMPO field of regard (r = 0.972, m = 0.986). Overall, we calculate a mean bias in our new TEMPO algorithm compared to the NASA standard product of only -0.05 x $10^{15}$ molecules cm$^{-2}$ (a normalized mean bias of -1.5 %)."*

Page 19, Line 19:

*"Our TEMPO algorithm also demonstrates good performance when evaluated against the stratospheric NO$_2$ columns provided with the NASA SPv3 standard product, but further independent evaluation using ground-based spectrometer network observations will be beneficial."*

**RC:** Page 6, line 5: a short justification is needed about the fact that data are restricted to SZA smaller than 80◦.

**AR:** In response to the reviewer's comment, we have added the following text to the manuscript:

Page 6, Line 5:

"We restrict all data to solar zenith angles smaller than 80° to avoid exceedingly long path lengths."

**RC:** Page 7, line 1-4: Monthly mean of GOME-2 tropospheric NO2 columns is used as initial a-priori tropospheric NO2 estimate. How is it done in practice? Are the GOME-2 data first gridded on the same 0.1◦x0.1◦ regular grid as OMI? A clarification would be helpful here or at the end of the description of the GOME-2 data in Section 2. Also, since the tropospheric NO2 column can show strong diurnal changes, is the GOME-2 tropospheric column a good estimate of the column at the OMI overpass time?

**AR:** We have clarified agree with the reviewer that tropospheric NO2 can show strong diurnal changes. However, diurnal variability tends to be highest over NOx source regions, and smaller over non-source

regions. For example, Boersma et al. (2008) demonstrate in their comparison of SCIAMACHY and OMI pixels (roughly the same time differences as we would expect from GOME-2 and OMI in our case) that the global probability distribution of tropospheric NO2 over the Pacific Ocean at the two overpass times show only a small offset, and they attribute this to a negative bias from the OMI retrieval. The high diurnal variability over source regions is inconsequential – these regions should be masked out during our algorithm and should therefore introduce less impact. However, we agree with the reviewer that ideally independent observations from the appropriate time of day would be used.

In response to the reviewer's comment we have added the following text to our manuscript:

Page 7, Line 7:

*"The GOME-2 observations were filtered using recommended quality flags and retaining pixels with cloud radiance fraction less than 0.2, then gridded to the same resolution as our OMI grid."*

*"Ideally, an independent LEO tropospheric estimate for as close to the TEMPO observation time would be used. Nonetheless, diurnal variability in tropospheric NO2 columns outside of source regions tends to be small (Boersma et al. 2008), and in our case source regions are masked out in a later step."*

**RC:** Page 3, line 6: 'Richter et al., 2005' instead of 'Richter et al. 2005'. Similar corrections should be done on the same page at lines 7, 13, 14; on page 3, line 3; on page 4, line 20; on page 6, line 2.

**AR:** We have made these corrections.

**RC:** Page 4, line 19: 'available' instead of 'avialable'

**AR:** We have made this correction.

**RC:** Page 7, line 30: one bracket should be removed after '2013'.

**AR:** We have made this correction.

---

## Author Comment (AC2) · 17 Oct 2018

*Response to Referee #2*

We thank the reviewer for their comments, and respond to each below.

**Reviewer Comment (RC):** The paper describes the adaptation of a stratosphere/troposphere separation algorithm to the upcoming geostationary satellite instrument TEMPO. It is well written, logically structured and convincing in its conclusions. The paper should be published on AMT after dealing with the following issues:

**Author Response (AR):** We thank the reviewer very much for their positive and constructive remarks.

**RC:** Gridding approach: The authors perform a gridding as very first step (page 5, line 29). This is not optimal, as satellite pixels with potentially very different conditions (i.e. a low total column over a clouded pixel next to a high total column over a power plant stack without clouds, both within the same 0.1◦ grid box) are just averaged, with consequences hard to foresee due to the many nonlinearities involved. I would like to encourage the authors to rethink this approach and go for a different order, i.e. applying the filter on Strop,prior and the masking of pixels high ratio of strat vs trop AMF on individual satellite pixels rather than averaged 0.1◦ grid pixels.

**AC:** We agree with the reviewer that averaging pixels before running the algorithm may introduce unknown effects, and we thank the reviewer for the opportunity to clarify our strategy. Indeed, the TEMPO algorithm should be performed on the individual TEMPO pixels. Nonetheless, testing our algorithm on the individual OMI pixels would not necessarily capture issues that will be unique to the TEMPO viewing geometry. For this reason, and given the absence of real TEMPO data, we have treated the individual gridded satellite pixels as a proxy for individual TEMPO pixels, and focus in this manuscript on the performance of the algorithm with respect to the limited field of regard.

We recommend that the operational algorithm be performed on the individual TEMPO pixels once they are available. We thank the reviewer for bringing up this point, and have included the following text and clarifications in our manuscript:

Page 6, line 27:

*"Although we begin our implementation with the OMI observations gridded to 0.1 x 0.1, the TEMPO algorithm would be performed on individual TEMPO pixels. In other words, here we are treating our gridded OMI observations as TEMPO pixels."*

**RC:** Tropospheric columns: Please provide some information on the frequency distribution of tropospheric columns over remote regions. Do negative columns occur? How large and how variable is the tropospheric column at the edges of the TEMPO domain?

**AR:** The reviewer asks a good question regarding potential negative tropospheric columns, and a related question about the columns at the TEMPO edges. While it is relatively difficult to identify "remote"

regions within this field of regard, we will treat the pixels immediately adjacent to the western TEMPO edge as "remote" for this investigation.

Figure R1 shows the histogram of the tropospheric NO2 columns that result from our TEMPO algorithm for the pixels directly adjacent to the western TEMPO edge (10 pixels deep) on July 15, 2007. We also show the cumulative probability distribution for the tropospheric NO2 columns along this region. For comparison, we include the tropospheric NO2 columns for the identical pixel locations from the OMI v3.0 standard product retrieval.

[Figure]

As you can see, our algorithm indeed results in negative tropospheric columns. The distribution is consistent with the independent SPv3 standard product retrieval for the same pixels. In both algorithms, about 37% of the pixels along this region are negative, as we would expect for a noisy signal close to zero. The mean tropospheric NO2 column along this western edge in our TEMPO algorithm is $0.71 \times 10^{14}$ +/- $3.63 \times 10^{14}$ molecules cm$^{-2}$, consistent with the mean tropospheric column in the same pixels from the standard OMI product is $0.98$ +/- $3.38 \times 10^{14}$ molecules cm$^{-2}$.

In summary, to answer the reviewer's questions: we find negative tropospheric columns in our algorithm, and these are consistent with the distribution from the independent standard product retrieval from NASA. These distributions also answer the reviewer's question about the magnitude and variability of the tropospheric column along the edge: we calculate a mean of $0.71 \times 10^{14}$ molecules cm$^{-2}$, with a standard deviation of $3.63 \times 10^{14}$ molecules cm$^{-2}$.

In response to the reviewer's question, we have added the following material to our manuscript:

Page 14, line 11:

*"We further evaluate the performance of our algorithm by comparing the $NO_2$ tropospheric column distribution along the western-most edge (1˚ deep) of the TEMPO field of regard with the $NO_2$ tropospheric column distribution resulting from the independent NASA SPv3 standard product. In this relatively remote region of the field of regard, we find a similar mean and standard deviation in column density ($0.71 \times 10^{14}$ +/- $3.63 \times 10^{14}$ molecules cm$^{-2}$ in our TEMPO algorithm and $0.98$ +/- $3.38 \times 10^{14}$ molecules cm$^{-1}$ in the NASA SPv3). The fraction of negative columns that are observed in our algorithm is consistent with the fraction of negative columns that occurs at the same location from the standard product (~37%)."*

**RC:** After introducing LEO on page 1, line 19, please use it (e.g. page 2, line 9; page 3, line 1).

**AR:** We have made the appropriate changes to the manuscript by replacing "low earth orbit" with "LEO" where applicable.

**RC:** Please comment which STS algorithm is foreseen for operational processing of TEMPO

**AR:** We thank the reviewer for providing the opportunity to clarify processing strategy. In response to this comment, we have added the following text to our manuscript:

Page 18, Line 9:

*"Given these results, our recommendation for TEMPO is to use a climatological estimate (e.g. a 30-day mean) of stratospheric NO2 for context outside of the TEMPO field of regard during near-real-time retrieval if LEO observations are unavailable. This climatological estimate can be constructed based on satellite-derived observations in LEO from the preceding year and corrected for the time of day based on model results or other independent observations. We would then propose a later re-processing of the data that incorporates the daily LEO observations when available from the correct observation day."*

---

## Author Response (AR2)

Re: Manuscript amt-2018-148, "Stratosphere-troposphere separation of nitrogen dioxide columns from the TEMPO geostationary satellite instrument" by Jeffrey A. Geddes, Randall V. Martin, Eric J. Bucsela, Chris L. McLinden, and Daniel Cunningham

Dear Handling Editor and Production Team,

Many thanks for your time and consideration of our manuscript. We are delighted that this has been accepted for publication.

We have corrected the typo you identified. In addition, we further identified and corrected the following typos (mainly formatting issues) while preparing our manuscript for production:

Page 7, Line 12: "… tropospheric $NO_2$ columns outside of source regions…" (We subscripted the "2" in "$NO_2$")

Page 13, Line 23: "… tropospheric column due to this issue…" (We inserted the word "to" between "due" and "this")

Page 14, Line 15: "…0.71 x $10^{14}$ ± 3.63 x $10^{14}$ molec $cm^{-2}$ in our…" (We superscripted "14" in "$10^{14}$")

Page 7, Line 8 now reads: "…pixels with cloud radiance fraction less than 0.5, then gridded…" (this originally said "cloud radiance fraction less than 0.2")

Thanks again for your time and consideration.

Jeffrey A. Geddes

jgeddes@bu.edu